# Seed Morphological Properties Related to Taxonomy in *Silene* L. Species

José Javier Martín-Gómez [1], José Luis Rodríguez-Lorenzo [2], Ana Juan [3], Ángel Tocino [4], Bohuslav Janousek [2] and Emilio Cervantes [1,*]

[1] Instituto de Recursos Naturales y Agrobiología (Consejo Superior de Investigaciones Científicas), 37008 Salamanca, Spain; jjavier.martin@irnasa.csic.es

[2] Plant Developmental Genetics, Institute of Biophysics v.v.i, Academy of Sciences of the Czech Republic, 612 65 Brno, Czech Republic; rodriguez@ibp.cz (J.L.R.-L.); janousek@ibp.cz (B.J.)

[3] Departamento de Ciencias Ambientales y Recursos Naturales, Universidad de Alicante, 03690 Alicante, Spain; ana.juan@ua.es

[4] Departamento de Matemáticas, Facultad de Ciencias, Universidad de Salamanca, 37008 Salamanca, Spain; bacon@usal.es

[*] Correspondence: emilio.cervantes@irnasa.csic.es

**Abstract:** *Silene* taxonomy, traditionally based on morphological characteristics, is now driven by DNA sequence analysis. While the usefulness of both morphological and molecular methods remains undisputed, there is an interest in the identification of the morphological characteristics useful in taxonomy. A quantitative morphological analysis of seeds belonging to *Silene* species is presented here and is based on seed image samples for 95 populations belonging to 52 species (49 species of *Silene* and 3 related species). According to the silhouette of lateral views of their seed images, *Silene* species are classified into three groups: smooth, rugose and echinate. The measurements taken for the lateral and dorsal views of the seeds include area, circularity, roundness, aspect ratio and solidity; differences between groups are found for all characteristics. Solidity is the ratio between the area of the seed silhouette and the corresponding convex hull. It is related to seed convexity and is the measurement with the lowest coefficient of variation. In the lateral views, solidity values are conserved, while in the dorsal views, differences are found between the three groups. The group of echinate seeds has the highest values of solidity in the dorsal views, and their species belong to *S.* subg. *Behenantha* and *S.* subg. *Lychnis*. The group of smooth seeds contains mainly species corresponding to *S.* subg. *Silene*, while species of *S.* subg. *Lychnis* are absent.

**Keywords:** caryophyllaceae; circularity; convexity; echinate seeds; roundness; rugose seeds; smooth seeds; solidity of seeds

## 1. Introduction

A significant proportion of species in the family Caryophyllaceae *Juss.* belong to the genus *Silene*. Their classification, originally based on morphological criteria [1,2], is now also based on the input of molecular approaches [3], but the integration of morphology and DNA sequence analysis requires an accurate definition of the characteristics used for the taxonomic descriptions. Characteristics that describe seed morphology may appear independently in separate lineages over the course of evolution (homoplasy) [4–9]. This hampers the application of morphology in the taxonomy of *Silene*. In contrast, synapomorphy refers to characteristics shared by sister taxa, with their common ancestor confirming the relationship between the shared phenotype and taxonomic position [3,4]. In this context, a re-consideration of the measurements traditionally used in morphology from the point of view of geometry may help in the description of seed shape and the identification of characteristics relevant for taxonomy.

Morphological descriptions of seeds have been applied for a long time in the taxonomy of the genus *Silene*. Given the complexity and broad geographic distribution of this genus, work has focused on a range of diverse flora, for example, those of Egypt [8], Iran [10–12], Jordan [13], Korea [14], Pakistan [15], Spain [16], the Mediterranean area [17,18] and Turkey [19]. In addition to dealing with different species according to the flora of each country, the characteristics that are the object of study are diverse and defined differently for each group. The availability of methods for the description of seed morphology based on simple morphological techniques may be a tool for the taxonomy of this genus [20–22].

In relation to general seed shape, descriptions lack qualitative measurements and are based on terminology, such as circular, globose, reniform and piriform. In previous work, we described and quantified seed shape by comparison with geometric models because *Silene* species present an interesting diversity of seed shapes. In their lateral views, they resemble cardioid or cardioid-derived figures [20,21], while a range of geometric figures may serve as models for the dorsal views [22]. For both the lateral and dorsal views of seeds, species were classified by their similarity to different geometric models defined by algebraic equations (LM and DM stand, respectively, for lateral and dorsal models). The shape quantification was given as a measurement, termed the *J* index, that expressed the percentage of similarity between a seed image and a model. The *J* index for the model LM1 (the cardioid) had values above 90 in 12 species of *Silene* subg. *Behenantha* [20,21]. These species were: *S. acutifolia* Link ex Rohrb., *S. conica* L., *S. diclinis* (Lag.) M. Laínz, *S. dioica* (L.) Clairv., *S. latifolia* Poir., *S. littorea* Brot., *S. noctiflora* L., *S. pendula* L., *S. uniflora* Roth., *S. viscosa* Pers., *S. vulgaris* (Moench) Garcke and *S. zawadskii* Fenzl). Moreover, the *J* index was above 90 for the model LM1 in seven species belonging to *S.* subg. *Silene*: *S. gallica* L., *S. italica* (L.) Pers., *S. nutans* L., *S. otites* Sm., *S. portensis* L., *S. saxifraga* L. and *S. vivianii* Steud. However, the seeds of *S. diclinis*, *S. latifolia* and *S. littorea* had higher *J* index values with models LM2 and LM4 than with the cardioid model, while *S. gallica* had the highest value with LM3 [20]. Other species had the highest scores with other models, such as *S. diversifolia* Otth. with LM7 and *S. nicaeensis* All. and *S. scabriflora* Brot., which both adjusted better to model LM8 [21]. The analysis of the dorsal views indicated an association between seed convexity and adjustment to the lateral models LM2 and LM4. Higher values of seed convexity in the dorsal views were associated with good adjustment to the lateral models LM2 and LM4, and this was characteristic of *Silene* species belonging to *S.* subg. *Behenantha*. In this work, we present data on the similarity of new species to various models and introduce the analysis of convexity and solidity in seed images.

Solidity is a measure of convexity and is based on the relationship between two areas. It is defined by the ratio of the area of an object to the area of the convex hull of the same object [23]:

$$Solidity = \frac{area\ of\ a\ figure}{area\ of\ the\ corresponding\ convex\ hull} \tag{1}$$

The convex hull of a plane figure is the smallest convex set that contains it [23]. Consequently, solidity is a quantitative measure and may oscillate between infinitesimal values and the unity. Convex figures are those for which both areas, i.e., the area of a region and the area of its convex hull, coincide. Therefore, convex figures are characterized by having a solidity value equal to one. For convex figures, the segment that joins any pair of its points is contained in the figure.

A convex plane curve is a plane curve that lies on one side of each of its tangent lines. A simple closed regular plane curve is convex if and only if its curvature has constant sign [24]. It corresponds to the border of a convex plane figure. In many instances, visual inspection of a geometric figure can give information about its convexity. For example, most of the seeds of the *Silene* genus show concavities in their lateral view [20,21], while some of them are convex in the dorsal view [22] (see Figure A1 in Appendix A for a graphical explanation of solidity).

In addition to the description of general seed shape, the morphological analysis of *Silene* seeds is often based on electron microscopy; there are detailed images representative

of many species and descriptions of characteristics [4,6,8,10–19]. In these approaches, the seeds are classified according to the shape of their cells or the morphological aspects of their boundaries. With respect to cell surface, the adjectives smooth, colliculate, echinate, mammalian, papillose and tuberculate are often used [4,13,15], but their precise descriptions are not always given.

One aim of this work is to present a summary or identity card for the seed morphology of 49 species of *Silene* and three species of two related genera (*Atocion* and *Viscaria*) based on 95 populations. Based on previous studies [20–22], we present data on similarities to geometric models for new, previously unstudied *Silene* species and closely related genera. This work also introduces a novel analysis of solidity in seed images in order to identify characteristics relevant for taxonomy purposes. In this work, we search for an easily applied classification method for seeds based on the silhouettes obtained from the analysis of photographic images. The obtained morphological data, based on geometric data and their comparison between species and taxonomical groups, would reveal the existence of a relevant source of information applicable to interpretations of taxonomic relationships.

## 2. Materials and Methods

### 2.1. Seeds Used in This Work

The seed stocks used in this work are listed in Appendix B (Table A1). They include from one to six stocks from different populations of the 49 species of *Silene*, as well as three samples belonging to the two small closely related genera *Atocion* (*A. rupestre* (L.) Oxelman) and *Viscaria* (*V. alpina* (L.) G. Don and *V. vulgaris* Röhl.), which sometimes have been classified as *Silene* [25]. The plant nomenclature (genera, subgenera, sections and species) and their authorities are adapted according to Plants of the World Online (POWO) [3,26].

### 2.2. Seed Images

Photographs of the lateral views of the seeds collected in Spain and Portugal were taken in Salamanca with a Nikon Stereomicroscope Model SMZ1500 equipped with a camera Nikon DS-Fi1 of 5.24 megapixels; dorsal views of these seeds were taken with a camera Nikon Z6 equipped with an objective AF-S Micro NIKKOR 60 mm f/2.8G ED. The remaining seeds were photographed in Brno with and Olympus SZX9 microscope coupled to an Olympus OM-D E-M10 II camera. Composed images containing 20 seeds per accession were prepared with Corel Photo Paint for the lateral and dorsal views of the seeds.

### 2.3. General Description of Size and Shape

Measurements of area (A), perimeter (P), length of the major axis (L), width (W), aspect ratio (AR is the ratio L/W), circularity (C), roundness (R) and solidity (S) were obtained for the lateral and dorsal views of seeds of each population with ImageJ [27]. A ruler was included in the photographs for the conversion of pixel units to length or surface units (mm or mm$^2$). The measurements of length and width are omitted from the results because the data for area and aspect ratio gave enough information concerning seed size. Circularity index is the ratio $(4\pi A)/P^2$ [28], while roundness is $(4A)/\pi L^2$ [29]. Seed surface irregularities increase the perimeter and reduce the values of circularity, leaving roundness unaffected. On the other side, elongated seeds have decreased roundness values.

### 2.4. Seed Shape Quantification and Testing of the Models: J Index

The *J* index indicates the similarity between a bi-dimensional image of a seed and the corresponding model. The *J* index is the percent of area that is shared by both images once they are superimposed for maximum similarity (Figure 1). It is quantified as:

$$J = S/T \times 100 \qquad (2)$$

where *S* is the area shared between the seed and the model, and *T* is the total area occupied by both images. The *J* index has a maximum value of 100, corresponding to the cases

where the geometric model and the seed image areas coincide. *J* index values given for each species correspond to the mean values obtained for 20 seeds. Higher scores indicate the similarity of a seed image with a given model, meaning that the model provides a precise definition of seed shape for a particular species. A good adjustment to the model is considered when the *J* index values were superior to 90 [20,30].

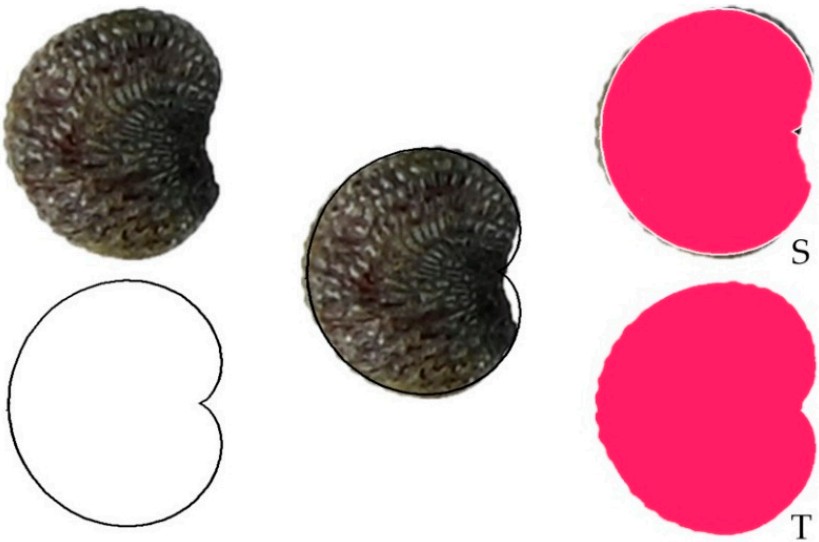

**Figure 1. Left**: Seed image (above) and geometric model (below). **Center**: The images of the seed and the model are superimposed. **Right**, in red: Shared area between the seed and the model (above) and total area (below). S = Shared area. T = Total area.

The data of the *J* index values for 31 *Silene* species were taken from previously available references [20–22], whereas for 17 *Silene* species (*S. acaulis*, *S. apetala*, *S. argaea*, *S. behen*, *S. borderei*, *S. caroliniana*, *S. chalcedonica*, *S. colorata*, *S. colpophylla*, *S. coronaria*, *S. elisabethae*, *S. foetida*, *S. flos-cuculi*, *S. flos-jovis*, *S. littorea*, *S. secundiflora* and *S. wolgensis*), the calculations were carried out as described [20,21] and using the above-mentioned calculation to find the best *J* index.

### 2.5. Statistical Analysis

The mean, minimum and maximum values and the standard deviation were obtained for all the measurements indicated above (A, P, L, W, AR, C and R), as well as for the *J* index with the different models. Statistics were performed on IBM SPSS statistics v28 (SPSS 2021) and R software v. 4.1.2 [31]. As some of the populations did not follow a normal distribution, non-parametric tests were applied for the comparison of populations. The Kruskal–Wallis test was used in the cases involving three or more populations, followed by stepwise stepdown comparisons by the ad hoc procedure developed by Campbell and Skillings [32]. P values inferior to 0.05 were considered significant. The coefficient of variation was calculated as CV = standard deviation/mean $\times$ 100 [33].

### 3. Results

#### 3.1. General Morphological Analysis

Representative photographs of the lateral and dorsal views of the seeds for each species are shown in Appendix A (Figure A2). The raw data used in the analysis are stored in Zenodo (https://zenodo.org/deposit/6772881; accessed on 15 June 2022). These include the values of the area, perimeter, length, width, circularity, roundness, aspect ratio and solidity for 95 populations belonging to 52 species (49 species of *Silene* and 3 related species). Table 1 contains a summary of the data.

**Table 1.** Summary of the values obtained for the different geometric parameters from the seeds of *Silene* in this study.

| | Dorsal View | Lateral View |
|---|---|---|
| Area | Mean value: 0.71 mm$^2$ (SD = 0.38; CV = 53.7)<br>Min. value: 0.24 mm$^2$ (*S. ramosissima*)<br>Max. value: 1.81 mm$^2$ (*S. pseudoatocion*) | Mean value: 0.92 mm$^2$ (SD = 0.50; CV = 53.9)<br>Min. value: 0.25 mm$^2$ (*S. disticha*)<br>Max. value: 2.94 mm$^2$ (*S. secundiflora*) |
| Perimeter | Mean value: 3.63 mm (SD = 0.99; CV = 27.3)<br>Min. value: 2.08 mm (*S. ramosissima*)<br>Max. value:6.49 mm (*S. secundiflora*) | Mean value: 3.90 mm (SD = 1.09; CV = 27.9)<br>Min. value: 2.13 mm (*S. disticha*)<br>Max. value:6.75 mm (*S. secundiflora*) |
| Circularity | Mean value: was 0.65 (SD = 0.11; CV = 17.1)<br>Min. value: 0.39 (*S. secundiflora*)<br>Max. value: 0.82 (*S. littorea*) | Mean value: 0.72 (SD = 0.08; CV = 10.9)<br>Min. value: 0.45 (*S. behen*)<br>Max. value: 0.85 (*S. littorea*) |
| Roundness | Mean value: 0.63 (SD = 0.14; CV = 21.7)<br>Min. value: 0.30 (*S. vivianii*)<br>Max. value: 0.89 (*S. disticha*) | Mean value: 0.82 (SD = 0.06; CV = 6.9)<br>Min. value: 0.73 (*S. mellifera*)<br>Max. value: 0.91 (*S. secundiflora*) |
| Aspect ratio | Mean value: 1.68 (SD = 0.44; CV = 26.1)<br>Min. value: 1.12 (*S. disticha*)<br>Max. value: 3.33 (*S. vivianii*) | Mean value: 1.22 (SD = 0.09; CV = 7.3)<br>Min. value: 1.11 (*S. secundiflora*)<br>Max. value: 1.40 (*S. mellifera*) |
| Solidity | Mean value: 0.939 (SD = 0.04; CV = 4.5)<br>Min. value: 0.822 (*S. secundiflora*)<br>Max. value: 0.987 (*S. littorea*) | Mean value: 0.957 (SD = 0.01; CV = 1.4)<br>Min. value: 0.917 (*S. behen*)<br>Max. value: 0.983 (*S. littorea*) |

SD: Standard Deviation; CV: Coefficient of Variation.

### 3.1.1. Area

- Dorsal view

  The mean value for the area in the dorsal view was 0.71 mm$^2$ (SD = 0.38; CV = 53.7). Area values in the dorsal views were between 0.24 mm$^2$ (*S. ramosissima*) and 1.81 mm$^2$ (*S. pseudoatocion*).

- Lateral view

  The mean value for the area in the lateral view was 0.92 mm$^2$ (SD = 0.50; CV = 53.9). Area values in the lateral views were between 0.25 mm$^2$ (*S. disticha*) and 2.94 mm$^2$ (*S. secundiflora*).

### 3.1.2. Perimeter

- Dorsal view

  The mean value for the perimeter in the dorsal view was 3.63 mm (SD = 0.99; CV = 27.3). Perimeter values in the dorsal views were between 2.08 mm (*S. ramosissima*) and 6.49 mm (*S. secundiflora*).

- Lateral view

  The mean value for the perimeter in the lateral view was 3.90 mm (SD = 1.09; CV = 27.9). Perimeter values in the lateral views were between 2.13 mm (*S. disticha*) and 6.75 mm (*S. secundiflora*).

### 3.1.3. Circularity

- Dorsal view

  The mean value for the circularity in the dorsal view was 0.65 (SD = 0.11; CV = 17.1). Circularity values in the dorsal views were between 0.39 (*S. secundiflora*) and 0.82 (*S. littorea*).

- Lateral view

  The mean value for the circularity in the lateral view was 0.72 (SD = 0.08; CV = 10.9). Circularity values in the lateral views were between 0.45 (*S. behen*) and 0.85 (*S. littorea*).

### 3.1.4. Roundness

- Dorsal view

The mean value for roundness in the dorsal view was 0.63 (SD = 0.14; CV = 21.7). Roundness values in the dorsal views were between 0.30 (*S. vivianii*) and 0.89 (*S. disticha*).

- Lateral view

The mean value for roundness in the lateral view was 0.82 (SD = 0.06; CV = 6.9). Roundness values in the lateral views were between 0.73 (*S. mellifera*) and 0.91 (*S. secundiflora*).

### 3.1.5. Aspect Ratio

In general, the aspect ratio values had a trend opposed to roundness.

- Dorsal view

The mean value for the aspect ratio in the dorsal view was 1.68 (SD = 0.44; CV = 26.1). The aspect ratio values in the dorsal views were between 1.12 (*S. disticha*) and 3.33 (*S. vivianii*).

- Lateral view

The mean value for the aspect ratio in the lateral view was 1.22 (SD = 0.09; CV = 7.3). The aspect ratio values in the lateral views were between 1.11 (*S. secundiflora*) and 1.40 (*S. mellifera*).

### 3.1.6. Solidity

- Dorsal view

The mean value for solidity in the dorsal view was 0.939 (SD = 0.04; CV = 4.5). The values of solidity in the dorsal views were between 0.822 (*S. secundiflora*) and 0.987 (*S. littorea*).

- Lateral view

The mean value for solidity in the lateral view was 0.957 (SD = 0.01; CV = 1.4). The values of solidity in the lateral views were between 0.917 (*S. behen*) and 0.983 (*S. littorea*).

### 3.2. Classification of Silene Species in Groups Based on Their Seed Silhouette

Three groups were defined that contained smooth seeds with no superficial projections (Figures 2 and 3), seeds with rounded projections or rounded tubercles (named as rugose seeds; Figures 4 and 5) and seeds with most part of the silhouette highly projected with acute tubercles or spines (named as echinate seeds; Figures 6 and 7). The group of smooth seeds was formed by the following 12 species (Figures 2 and 3): *S. apetala, S. borderei, S. colorata, S. colpophylla, S. crassipes, S. diversifolia, S. littorea, S. micropetala, S. nicaeensis, S. ramosissima, S. secundiflora* and *S. vivianii.* The group of rugose seeds was formed by the following 20 species (Figures 4 and 5): *S. acaulis, S. acutifolia, S. argaea, S. conica, S. coronaria, S. disticha, S. elisabethae, S. inaperta, S. mellifera, S. muscipula, S. nocturna, S. otites, S. portensis, S. saxifraga, S. scabriflora, S. sclerocarpa, S. stricta, S. tridentata, S. wolgensis* and *S. zawadzkii.* The group of echinate seeds was formed by the following 17 species (Figures 6 and 7): *S. behen, S. caroliniana, S. chalcedonica, S. coutinhoi, S. diclinis, S. dioica, S. flos-cuculi, S. flos-jovis, S. foetida, S. gallica, S. latifolia, S. noctiflora, S. nutans, S. pendula, S. pseudoatocion, S. viscosa* and *S. vulgaris.*

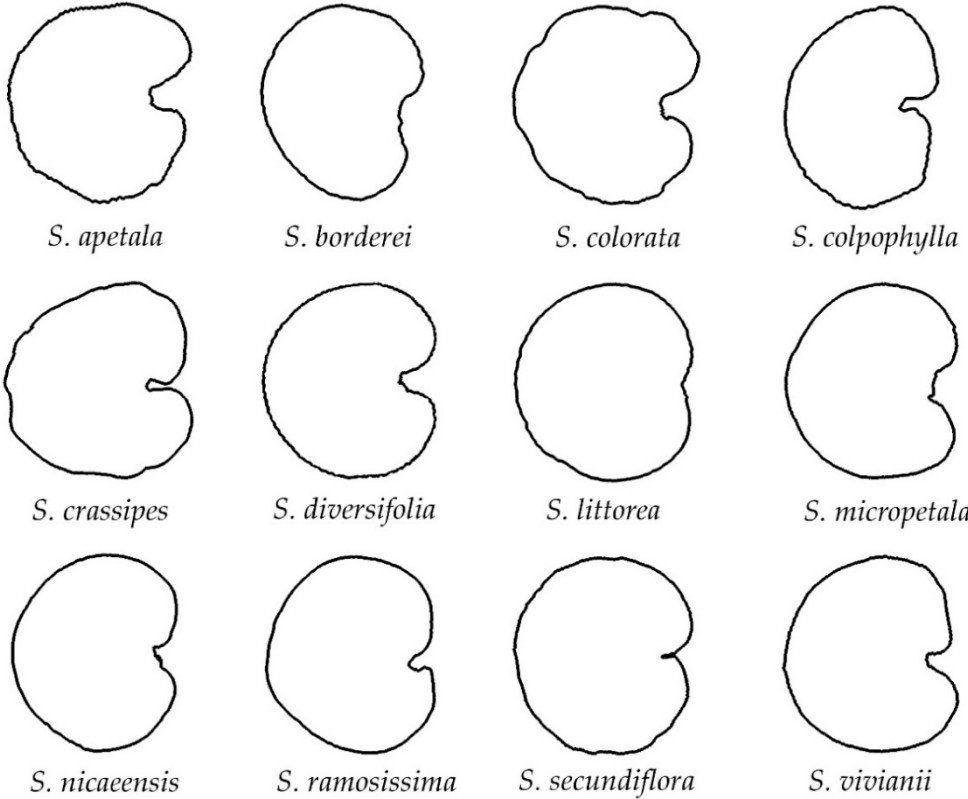

**Figure 2.** Silhouettes of the lateral views of *Silene* seeds belonging to the group of smooth seeds.

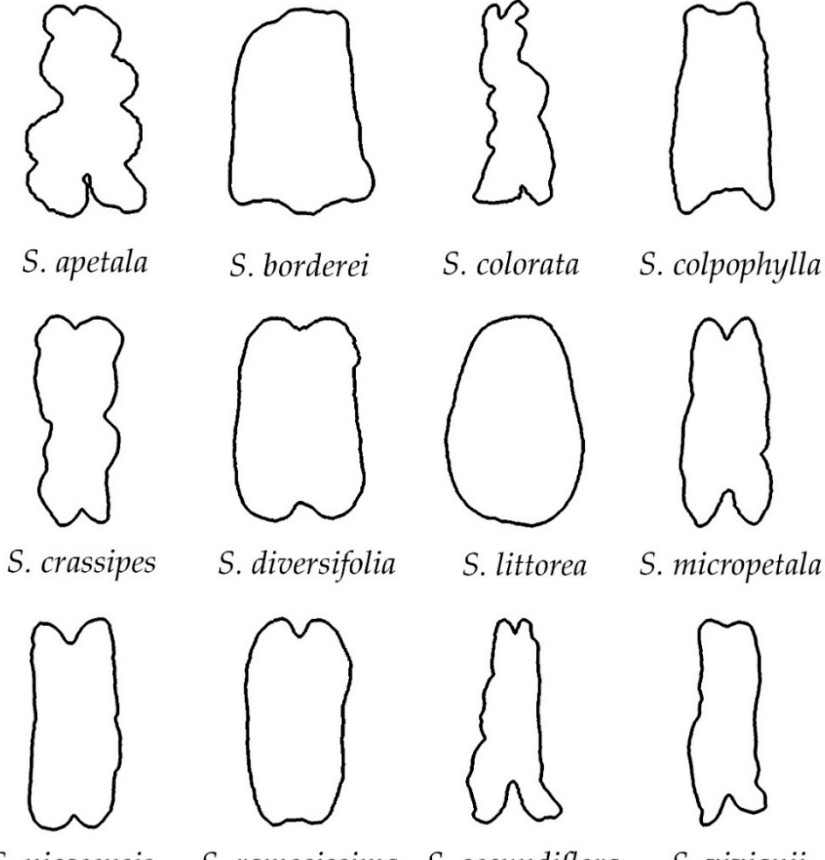

**Figure 3.** Silhouettes of the dorsal views of *Silene* seeds belonging to the group of smooth seeds.

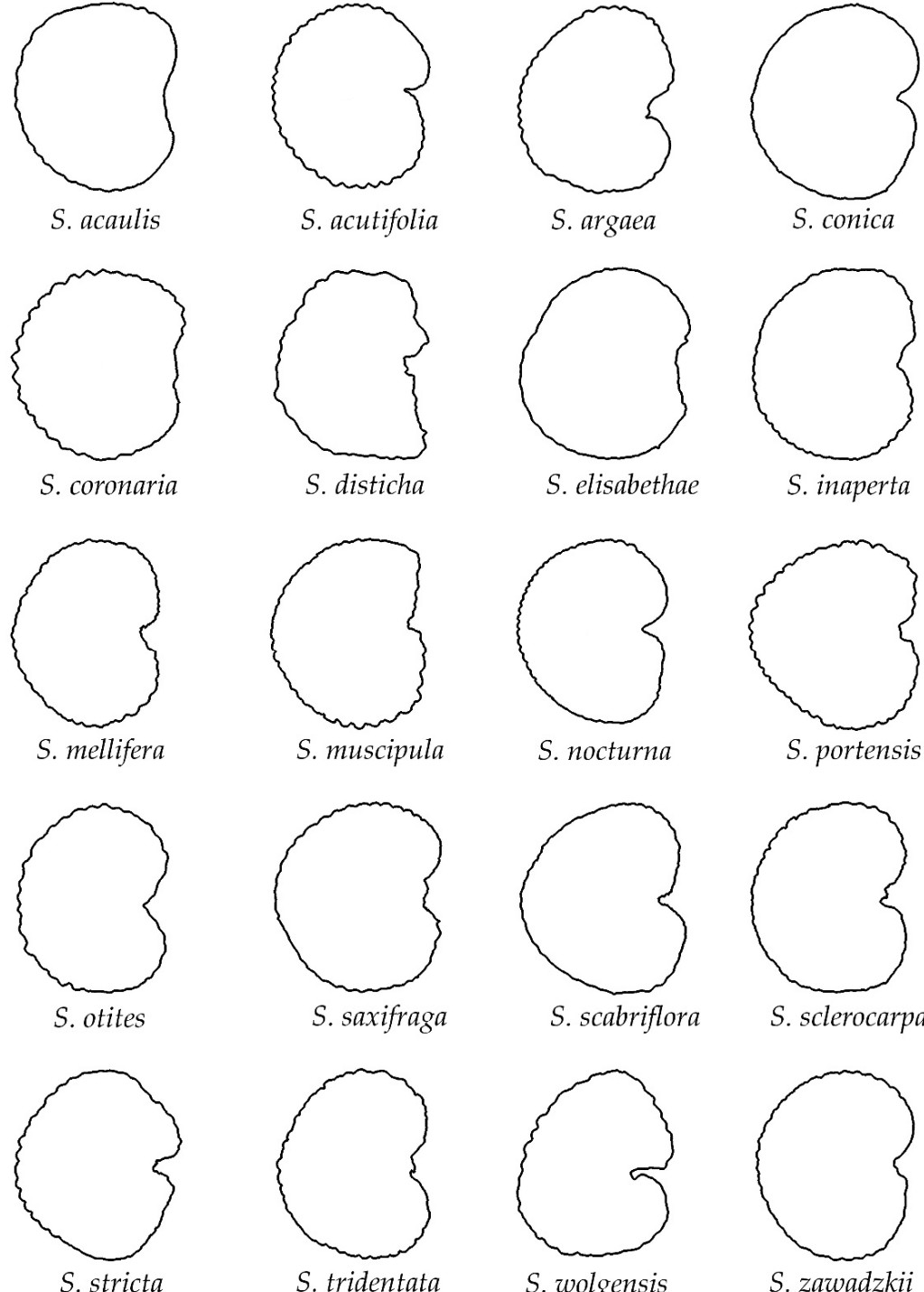

**Figure 4.** Silhouettes of the lateral views of *Silene* seeds belonging to the group of seeds with round tubercles.

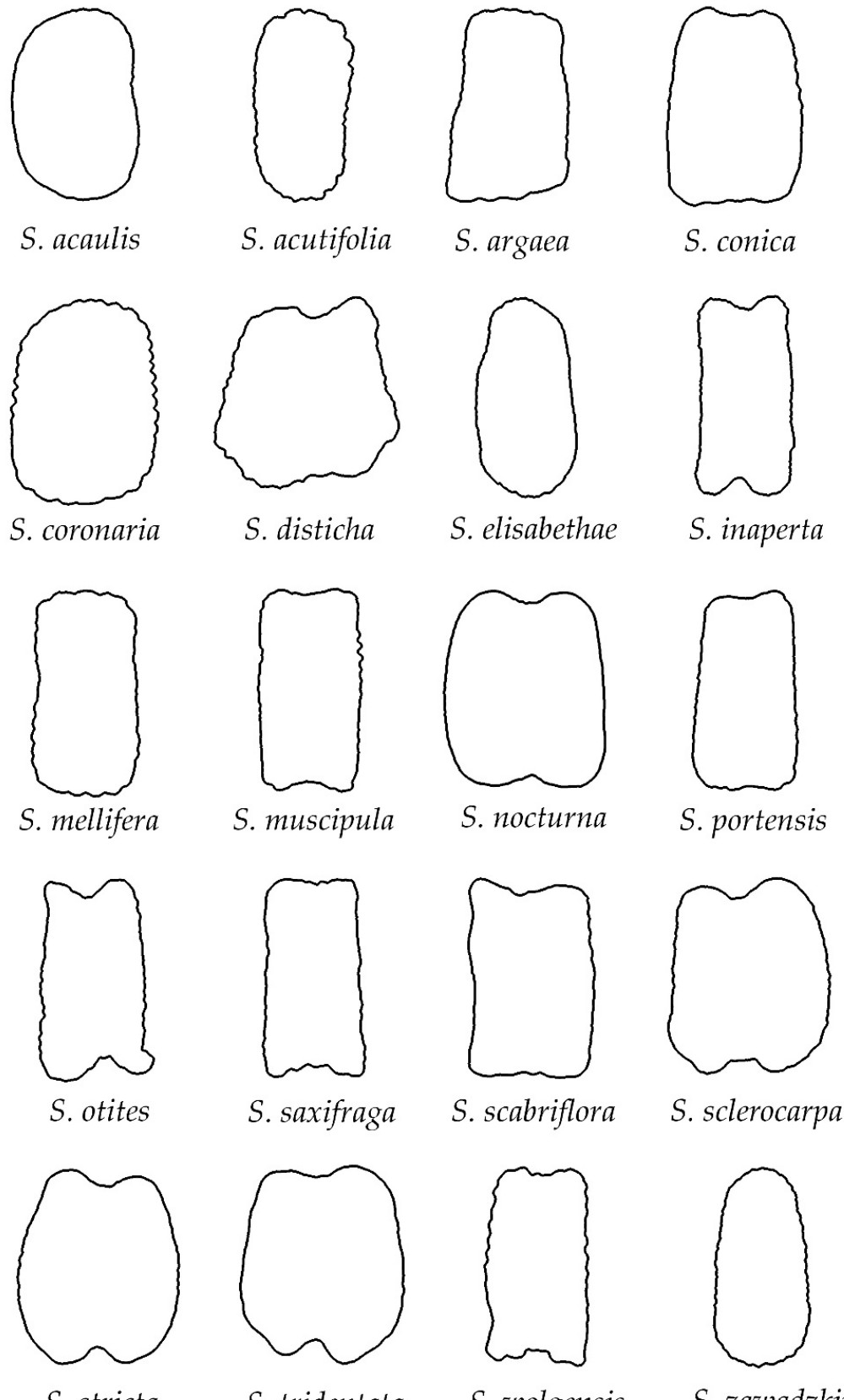

**Figure 5.** Silhouettes of the dorsal views of *Silene* seeds belonging to the group of seeds with round tubercles.

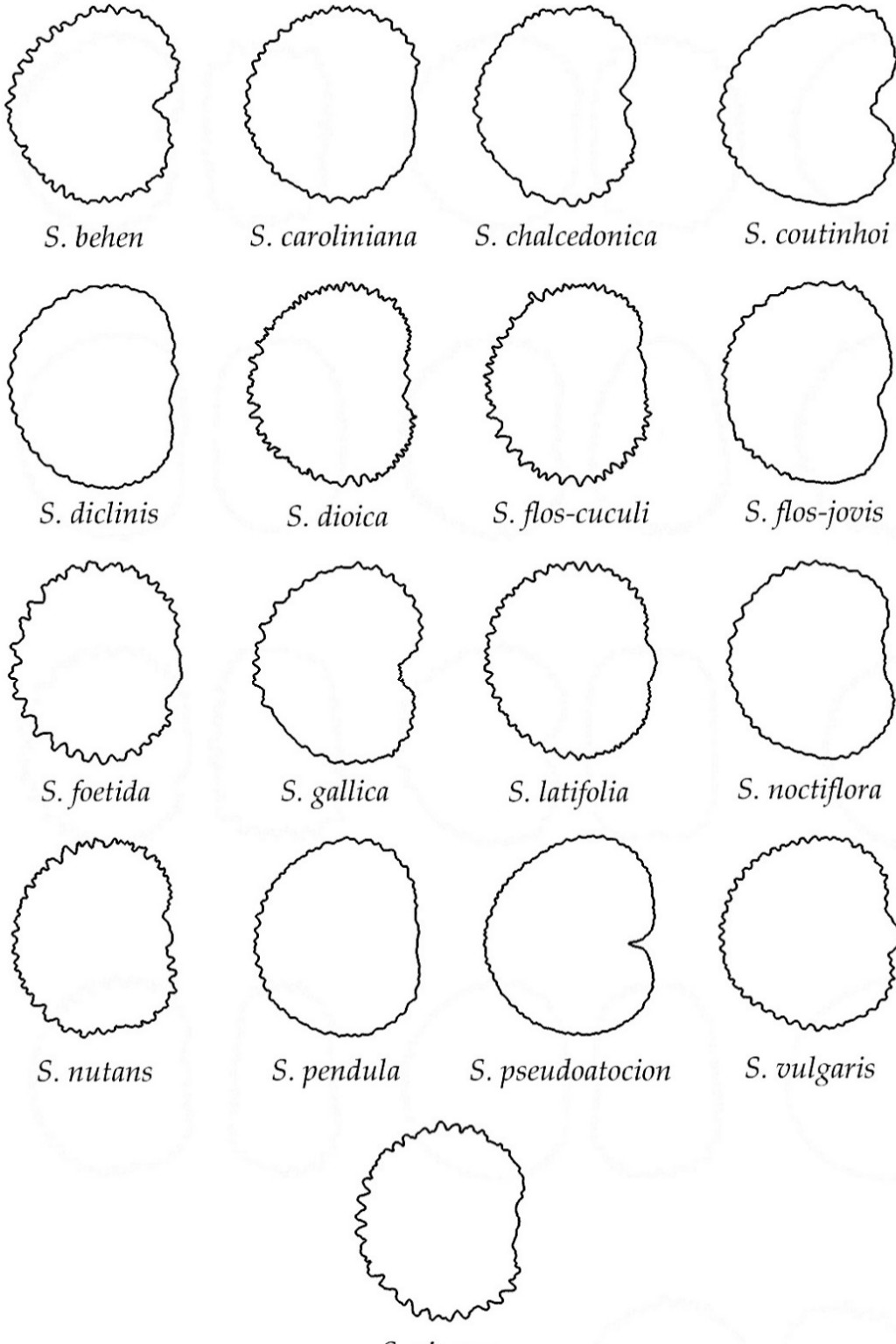

**Figure 6.** Lateral views of the silhouettes of *Silene* seeds belonging to the group of echinate seeds.

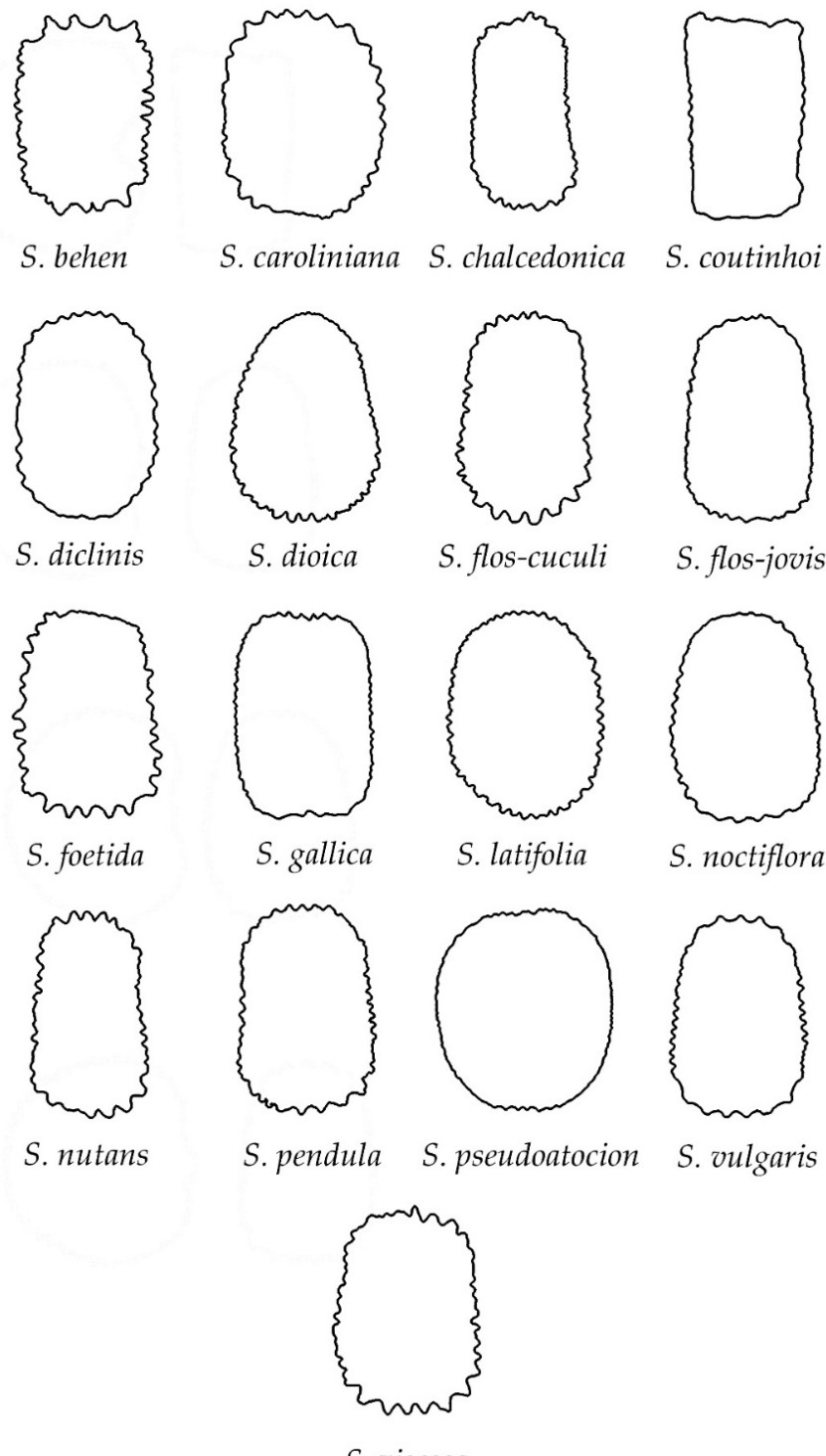

**Figure 7.** Silhouettes of the dorsal views of *Silene* seeds belonging to the group of echinate seeds.

The group of seeds with smooth surface contains a clear majority of species belonging to *S.* subg. *Silene* (11 out of 12, 91.7%) (specifically to the sections: *S.* sect. *Silene* (*S. apetala, S. borderei, S. colorata, S. diversifolia, S. micropetala, S. nicaeensis, S. secundiflora* and *S. vivianii*); *S.* sect. *Siphonomorpha* (*S. colpophylla* and *S. ramosissima*); and *S.* sect. *Lasiocalycinae* (*S. crassipes*)), a unique species belonging to *S.* subg. *Behenantha* (*S. littorea*), and no species of *S.* subg. *Lychnis* (Table 2, Figures 2 and 3). This seed group is mostly conformed by annual taxa (9 out of 12, 75%).

**Table 2.** Correlation between the three morphological groups of seeds and the taxonomical subgenera of *Silene*, according to [3].

| Morphological Type | Smooth | Rugose | Echinate |
|---|---|---|---|
| *S.* subg. *Silene* | *S. apetala*<br>*S. borderei*<br>*S. colorata*<br>*S. colpophylla*<br>*S. crassipes*<br>*S. diversifolia*<br>*S. micropetala*<br>*S. nicaeensis*<br>*S. ramosissima*<br>*S. secundiflora*<br>*S. vivianii* | *S. acaulis*<br>*S. argaea*<br>*S. disticha*<br>*S. inaperta*<br>*S. mellifera*<br>*S. muscipula*<br>*S. nocturna*<br>*S. otites*<br>*S. portensis*<br>*S. saxifraga*<br>*S. scabriflora*<br>*S. sclerocarpa*<br>*S. stricta*<br>*S. tridentata*<br>*S. wolgensis* | *S. coutinhoi*<br>*S. gallica*<br>*S. nutans*<br>*S. pseudoatocion* |
| *S.* subg. *Behenantha* | *S. littorea* | *S. acutifolia*<br>*S. conica*<br>*S. elisabethae*<br>*S. zawadzkii* | *S. behen*<br>*S. caroliniana*<br>*S. diclinis*<br>*S. dioica*<br>*S. foetida*<br>*S. latifolia*<br>*S. noctiflora*<br>*S. pendula*<br>*S. viscosa*<br>*S. vulgaris* |
| *S.* subg. *Lychnis* | | *S. coronaria* | *S. chalcedonica*<br>*S. flos-cuculi*<br>*S. flos-jovis* |

The dorsal view of the seeds revealed in many instances the existence of a remarkable channel or deep depression, giving them the presence of small or long irregular wings (Figure 3). Nevertheless, three types of dorsal morphologies can be identified: (i) truncated pyramid or truncated rectangular shapes with a shallow depression or channel in the upper and lower part of the seeds; (ii) truncated rectangular shape with noticeable deep depressions in both parts of the seeds and sinuate margins; and (iii) convex seeds without depressions. The first group was formed by *S. borderei*, *S. colpophylla*, *S. diversifolia*, *S. nicaeensis*, *S. ramosissima* and *S. viviannii*, whereas the second one grouped to *S. apetala*, *S. colorata*, *S. crassipes*, *S. micropetala* and *S. secundiflora*. Finally, the last group is represented by a unique species, *S. littorea*.

Similarly, the majority of the species included in the group of seeds with a rugose silhouette (Figure 4, Table 2) belong to *S.* subg. *Silene* (15 out of 20, 75%), mostly corresponding to *S.* sect. *Siphonomorpha* (*S. acaulis*, *S. mellifera*, *S. otites*, *S. saxifraga* and *S. wolgensis*) and *S.* sect. *Silene* (*S. nocturna*, *S. scabriflora*, *S. sclerocarpa* and *S. tridentata*). Other species in this group belong to other sections in *S.* subg. *Silene* such as *S. argaea* (*S.* sect. *Auriculatae*), *S. inaperta* and *S. stricta* (both *S.* sect. *Muscipula*). Only four species (20%) with rugose seeds belong to *S.* subg. *Behenantha*: *S. acutifolia*, *S. conica*, *S. elisabethae* and *S. zawadzkii*. Finally, *S. coronaria* is the only species with this type of seed belonging to *S.* subg. *Lychnis* (Table 2). Related to lifespan, this seed group was characterized by annual and perennial taxa (10 out of 20, 50% for both lifespan).

In relation to the dorsal view (Figure 5), the rugose seeds had either the shape of a isosceles trapezium or rectangle with a slight to moderate channel or depression in the upper and lower sides of the seeds (e.g., *S. conica*, *S. disticha*, *S. inaperta*, *S. muscipula*, *S. nocturna*, *S. otites*, *S. portensis*, *S. saxifraga*, *S. scabriflora*, *S. sclerocarpa*, *S. stricta*, *S. tridentata*,

*S. wolgensis*), or the seeds were convex without any channel or depression (e.g., *S. acaulis*, *S. acutifolia*, *S. argaea*, *S. coronaria*, *S. elisabethae*, *S. mellifera*, *S. zawadzkii*).

The group of seeds with echinate silhouette (Figure 6, Table 2) is mostly formed by 10 out of 17 species (59%) belonging to *S.* subg. *Behenantha* (*S. behen*, *S. pendula* and *S. vulgaris* in *S.* sect. *Behenantha*; *S. dioica*, *S. latifolia*, *S. diclinis* of *S.* sect. *Melandrium*; *S. caroliniana* and *S. viscosa* in *S.* sect. *Physiolychnis*; *S. foetida* in *S.* sect. *Acutifoliae*; and *S. noctiflora* in *S.* sect. *Elisanthe*) and 4 (23.5%) belonging to *S.* subg. *Silene* (*S. coutinhoi*, *S. gallica*, *S. nutans* and *S. pseudoatocion*). The subgenus *S.* subg. *Lychnis* is only represented by three species (17.5%): *S. chalcedonica*, *S. flos-cuculi* and *S. flos-jovis* (Table 2). Conversely to the previous seed groups, most the echinate seeds are perennial plants (12 out 17, 70.6%).

Conversely to the two previous groups of seeds, the dorsal view of the echinate seeds were predominantly convex (or rarely truncate rectangular, e.g., *S. coutinhoi*), characterized by the lack of a depression in both upper and lower part of the seeds and with a margin, in general, notably rugose or echinate (Figure 7).

### 3.3. Morphological Comparison between the Three Groups and Related Genera
### 3.3.1. Lateral View

Among the three groups of *Silene*, the highest area and perimeter were obtained for the group of echinate seeds whereas the lowest corresponded to the group of rugose seeds, which differences were statistically supported (Table 3). In the two species of *Viscaria*, the area and perimeter significantly showed the lowest values, while in *Atocion rupestre*, the area values were similar to the groups of echinate and smooth seeds ($p > 0.05$), but the perimeter was remarkably higher than any other group ($p < 0.05$) due to the length of the surface spines in this species. The aspect ratio also revealed significant differences among the three *Silene* groups, and the rugose and smooth seeds were characterized with the highest and lowest data, respectively. However, the species *A. rupestre* showed the highest values compared to any other studied group or species ($p < 0.05$), whereas *V. alpina* were quite similar to echinate and rugose seeds ($p > 0.05$). Related to the circularity, the rounded seeds showed the highest values, these being intermediate in smooth seeds and the lowest in the seeds with spines ($p < 0.05$). Comparing the *Atocion* and *Viscaria* species to the *Silene* groups, the values of circularity were the lowest in *Atocion rupestre* and the highest in *V. alpina* (both $p < 0.05$). The species *V. vulgaris* showed similar values to echinate seeds, but their differences were statistically supported. The results about roundness among the *Silene* groups revealed the highest values for the smooth seeds, intermediate in echinate seeds and the lowest in rugose seeds ($p < 0.05$). The species *A. rupestre* had the lowest roundness compared to any *Silene* group or *Viscaria* species. In the case of *V. alpina*, the roundness values were similar to rugose and echinate *Silene* groups ($p > 0.05$), and for *V. vulgaris*, the values were equivalent to the smooth *Silene* group ($p > 0.05$), characterized both with the highest values ($p < 0.05$). Finally, and related to solidity, the obtained data for the three *Silene* groups were quite similar among them (0.955–0.958, $p > 0.05$). For *Atocion rupestre*, the solidity was significantly lower compared to any other group or species. The two species of *Viscaria* showed closer values to *Silene*, but the species *V. alpina* showed the highest values ($p < 0.05$).

**Table 3.** Results of Kruskal–Wallis and post hoc tests for the lateral views of seeds in three groups of *Silene* species (smooth, rugose and echinate) and for species *Atocion rupestre*, *Viscaria alpina* and *Viscaria vulgaris*. Mean values and coefficients of variation (given in parentheses) are indicated for area (A), perimeter (P), aspect ratio (AR), circularity (C), roundness (R) and solidity (S). Values marked with the same superscript letter in each column correspond to populations that do not differ significantly at *p* < 0.05 (Campbell and Skillings's test). N indicates the number of seeds analyzed.

| Seed Type or Species | N | A | P | AR | C | R | S |
|---|---|---|---|---|---|---|---|
| Smooth | 361 | 1.08 [d] (52.1) | 4.14 [c] (23.0) | 1.18 [a] (7.2) | 0.75 [d] (6.4) | 0.85 [d] (6.7) | 0.958 [c] (1.2) |
| Rugose | 774 | 0.63 [c] (58.1) | 3.15 [b] (27.7) | 1.24 [d] (7.4) | 0.76 [e] (7.1) | 0.81 [b] (7.0) | 0.957 [c] (1.3) |
| Echinate | 770 | 1.14 [e] (37.8) | 4.56 [d] (18.7) | 1.23 [c] (6.9) | 0.67 [c] (12.3) | 0.82 [c] (6.3) | 0.955 [c] (1.5) |
| *A. rupestre* (*S. alpestris*) | 20 | 1.03 [de] (9.2) | 8.51 [e] (17.1) | 1.31 [e] (7.0) | 0.19 [a] (32.6) | 0.77 [a] (7.0) | 0.785 [a] (5.6) |
| *V. alpina* (*S. suecica*) | 20 | 0.33 [b] (10.6) | 2.31 [a] (5.9) | 1.23 [cd] (3.9) | 0.79 [f] (2.1) | 0.82 [bc] (4.0) | 0.970 [d] (0.8) |
| *V. vulgaris* (*S. viscaria*) | 60 | 0.26 [a] (14.0) | 2.25 [a] (8.0) | 1.19 [b] (5.6) | 0.65 [b] (7.1) | 0.84 [d] (5.8) | 0.947 [b] (1.0) |

### 3.3.2. Dorsal View

Among the groups of *Silene*, the values of area, perimeter, roundness and solidity values were significantly higher in the group of echinate seeds than in the other two groups (Table 4). Comparing between the smooth and rugose seeds, the area and perimeter were higher in the group of smooth seeds (*p* < 0.05), while the roundness and solidity values were higher in the group of rugose seeds (*p* < 0.05). Related to the two species of *Viscaria*, the area and perimeter showed the lowest values compared to the *Silene* groups and the species *A. rupestre* (*p* < 0.05). The area values of this latter species were similar to the smooth and rugose *Silene* groups (*p* > 0.05), but the data for the perimeter were significantly higher than for any of the *Silene* groups. For the roundness and solidity, this same species, *A. rupestre*, had the lowest values (*p* < 0.05) compared to any studied group. Conversely, the roundness values for *V. alpina* were equivalent to the group of smooth seeds in *Silene* (*p* > 0.05) and similar to echinate seeds for the solidity (*p* > 0.05). In the case of *V. vulgaris*, these values of roundness and solidity were intermediate between the smooth and rugose *Silene* seeds (*p* < 0.05).

**Table 4.** Results of Kruskal–Wallis and post hoc tests for the dorsal views of seeds in the three groups of *Silene* species (smooth, rugose and echinate) and for species *Atocion rupestre*, *Viscaria alpina* and *Viscaria vulgaris*. Mean values and coefficients of variation (given in parentheses) are indicated for area (A), perimeter (P), aspect ratio (AR), circularity (C), roundness (R) and solidity (S). Values marked with the same superscript letter in each column correspond to populations that do not differ significantly at *p* < 0.05 (Campbell and Skillings's test). N indicates the number of seeds analyzed.

| Seed Type or Species | N | A | P | AR | C | R | S |
|---|---|---|---|---|---|---|---|
| Smooth | 349 | 0.57 [d] (49.8) | 3.59 [c] (28.9) | 2.21 [d] (24.4) | 0.56 [b] (22.7) | 0.48 [b] (22.6) | 0.894 [b] (6.4) |
| Rugose | 742 | 0.52 [c] (43.2) | 3.00 [b] (21.1) | 1.62 [b] (23.6) | 0.70 [d] (11.4) | 0.65 [d] (22.5) | 0.945 [d] (3.4) |
| Echinate | 743 | 0.97 [e] (40.7) | 4.28 [d] (19.2) | 1.50 [a] (11.6) | 0.64 [c] (15.6) | 0.67 [e] (11.1) | 0.955 [e] (2.4) |
| *A. rupestre* (*S. alpestris*) | 15 | 0.51 [cd] (18.5) | 5.39 [e] (20.2) | 3.01 [e] (30.9) | 0.23 [a] (22.5) | 0.36 [a] (26.0) | 0.772 [a] (5.8) |
| *V. alpina* (*S. suecica*) | 20 | 0.23 [b] (16.1) | 2.11 [a] (9.7) | 2.01 [d] (8.0) | 0.65 [c] (5.9) | 0.50 [b] (7.9) | 0.963 [e] (1.2) |
| *V. vulgaris* (*S. viscaria*) | 60 | 0.19 [a] (15.9) | 2.02 [a] (7.6) | 1.80 [c] (7.3) | 0.58 [b] (8.7) | 0.56 [c] (7.3) | 0.931 [c] (1.9) |

Among the three groups of *Silene* (Table 4), circularity was higher in the rugose seeds, intermediate in echinate seeds, and lower in the group of smooth seeds (*p* < 0.05). The lowest values of circularity were found for the *A. rupestre*, whereas the values for the two species of *Viscaria* were similar to the group of smooth seeds (*V. vulgaris*, *p* > 0.05) and to the group of echinate seeds (*V. alpina*, *p* > 0.05).

The highest values of the aspect ratio among the *Silene* groups were identified for the smooth seeds of *Silene* (Table 4), whereas the echinate seeds showed the lowest values. Related to *Atocion* and *Viscaria* taxa, the species *A. rupestre* showed the highest aspect

ratio values (*p* < 0.05), whereas these data were intermediate for *V. alpina* and *V. vulgaris* compared to the *Silene* groups of smooth and rugose seeds

The results of solidity in the dorsal views were different in each of the three groups of *Silene*, with highest values in the group of echinate seeds, intermediate in rugose seeds, and smallest values in smooth seeds, due to their pronounced channel (dorso canaliculata). *V. alpina* had values similar to the group of echinate seeds, *V. vulgaris* had values comprised between those obtained for the groups of smooth and rugose seeds. The values of solidity in *A. rupestre* were lower than those obtained in the groups formed with the species of *Silene*.

### 3.4. Analysis of Shape by Geometric Models in Each of the Three Groups of Silene

The values given are taken from references [19–21] or belong to this work when no reference is given.

#### 3.4.1. Smooth Seeds

The seeds of species included in the group of smooth seeds corresponded to a broad range of lateral models (LM): LM1 (*S. littorea*, with a *J* index value of 91.4, Figure 8; *S. secundiflora*, 91.1, Figure 8; *S. vivianii*, 90.5 [21]), LM5 (*S. diversifolia*, 91.0 [21]), LM7 (*S. colorata*, 90.2, Figure 8; *S. ramosissima* 91.9 [22]), LM8 (*S. nicaeensis*, 90.1 [21]) or are difficult to attribute to any of the models (*S. apetala, S. borderei, S. colpophylla, S. crassipes* [21], *S. micropetala* [21]).

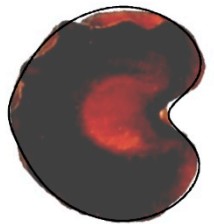 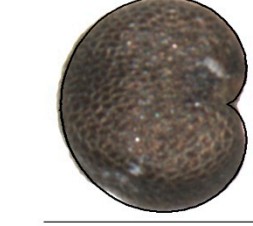 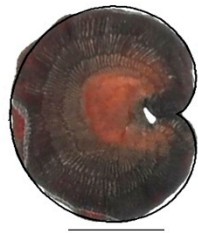

*S. colorata* vs. LM7  *S. littorea* vs. LM1  *S. secundiflora* vs. LM1

**Figure 8.** Representative images of the group of smooth seeds. The mean values of *J* index obtained with the models shown are 90.2 (*S. colorata* with LM7); 91.4 (*S. littorea* with LM1) and 91.1 (*S. secundiflora* with LM1).

#### 3.4.2. Rugose Seeds

The species included in this group correspond to the lateral models LM1, LM2, LM4, LM5 and LM8. High *J* index values with LM1 were obtained in *S. acutifolia*, 91.2 [20]; *S. conica*, 92.5 [21]; *S. elisabethae*, 90.1 (Figure 9); *S. mellifera* 89.8 [20]; *S. otites*, 90.7 [20], *S. portensis*, 90.8 [21], *S. saxifraga*, 90.3 [20] and *S. zawadzkii*, 90.0 [20]); with LM2 (*S. coronaria*, 93.5, Figure 9); LM4 (*S. acaulis*, 91.0, Figure 9); LM5 (*S. tridentata* 90.8 [21]); LM8 (*S. inaperta*, 91.0 [22]; *S. scabriflora*, 91.3 [21]). The following species remain undefined in relation to the lateral model: *S. argaea, S. disticha, S. muscipula, S. nocturna, S. sclerocarpa, S. stricta* and *S. wolgensis*.

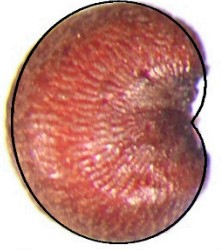 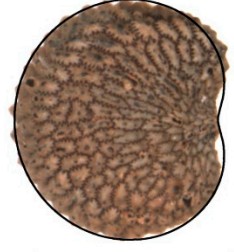 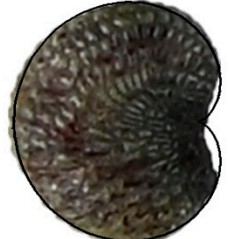

*S. acaulis* vs. LM4  *S. coronaria* vs. LM2  *S. elisabethae* vs. LM1

**Figure 9.** Representative images of the group of rugose seeds. The mean values of *J* index obtained with the models shown are of 91.0 for *S. acaulis* with LM4; 93.5 for *S. coronaria* with LM1 and 90.1 for *S. elisabethae* with LM1.

### 3.4.3. Echinate Seeds

Echinate seeds adjusted well with models LM1 (*S. behen*, 90.3, Figure 10; *S. dioica*, 91.6 [20]; *S. gallica*, 90.0 [20]; *S. latifolia*, 92.7 [20]; *S. noctiflora*, 93.6 [20]; *S. nutans*, 91.6 [20]; *S. pendula*, 90.8 [20], *S. viscosa*, 92.6 [20]; *S. vulgaris*, 91.2 [20]), LM2 (*S. caroliniana*, 90.4, Figure 10; *S. foetida*, 90.2, Figure 10) and LM4 (*S. diclinis*, 91.5 [20]; *S. flos-jovis*, 90.1, Figure 10). Related to the lateral models, *S. chalcedonica*, *S. coutinhoi*, and *S. flos-cuculi* remain undefined in this group.

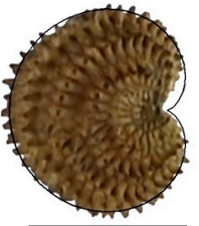 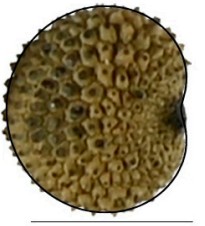 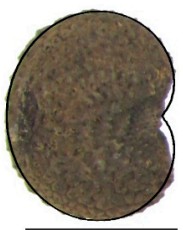 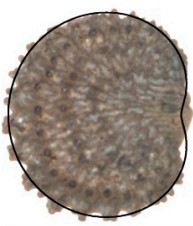

*S. behen* vs. LM1     *S. caroliniana* vs. LM2     *S. flos-jovis* vs. LM4     *S. foetida* vs. LM2

**Figure 10.** Representative images of the group of echinate seeds. The mean *J* index values obtained with the models shown are 90.3 for *S. behen* with LM1; 90.4 for *S. carolininaa* with LM2; 90.1 for *S. flos-jovis* with LM4; and 90.2 for *S. foetida* with LM2.

## 4. Discussion

The notable seed morphology of *Silene* is worth of attention to be investigated to give more evidences for the infrageneric classification of this and close-related genera. Previous studies about morphological qualitative characters of the *Silene* seeds were mainly based on the surface appearance of the cells [4,6,15] and the presence or lack of wings [4,15], although other seed features were also included, such as the testa cell types and the outline shape or the state of the hilum [4]. Forty-nine species of *Silene* were classified here in three groups according to the silhouettes obtained for the lateral face of their seeds, here named as smooth, rugose and echinate. Conversely to Hoseini et al. [4], this classification was based on the presence or lack of different types of projections (tubercles, spines) on the outline of the lateral face of the seeds but not on particular morphological aspects of cells in the seed surface [4]. The classification proposed here revealed a clear definition for each group, and thus, the species were classified exclusively in one of the described seed groups. The infrageneric classification of *Silene*, at the taxonomic rank of subgenus, seemed to be somehow supported by the type of seed silhouette, since smooth and echinate seeds were clearly dominant in *S.* subg. *Silene* and *S.* subg. *Behenantha*, respectively. In addition, the seeds of the close-related species *Atocion rupestre*, *Viscaria vulgaris* and *V. alpina* have been analysed as external controls to observe the variations in the characteristics studied, which showed, in general, a higher relationship between the three groups of *Silene* than with any of these external groups.

The classification in these three groups was consistent with other morphological seed features. These groups corresponded to different geometric models based on the mathematical formula [20,21], though the lateral model LM1 was shared by the three seed groups. A high variety of models describes the lateral seed shape of rugose seeds, but the group of echinate seeds was characterized by good adjustments with the lateral models LM1, LM2 and LM4 [20,21] and DM1 to DM4 in the dorsal views [22]. In addition, the application of these geometric models would give a quantitative system to classify the outline shape of the seeds [20–22,30], helping to give a more accurate definition of the qualitative description of the seed morphology widely used in the literature e.g., [4,6,7,15]. Differences between these three groups were also found for the majority of quantitative characteristics analysed, both in the lateral and the dorsal views, including area, perimeter, circularity and roundness. In most cases, differences were found for all possible pair combinations between groups, such for area, perimeter, circularity and roundness in the lateral seed views and for perimeter, circularity and roundness in the dorsal seed views.

The case of the calculation of the solidity for *Silene* presents two characteristics. First, it is the most conserved measurement based on (1) the lowest coefficient of variation values and (2) there being no differences in the lateral view between the groups. Solidity is a measure of convexity based on the relation between two plane figures [23]: the seed image and the corresponding convex hull. Although in the lateral views, there were no significant differences in convexity between the groups, in the dorsal views, the solidity values were significant higher in the group of seeds with spines (echinate seeds). This is somehow counter-intuitive because one would expect higher solidity values in seeds with smooth surface, as happens for the species *S. littorea*, which gave the maximum values of solidity both in the lateral and dorsal views (the unique exception within the smooth seed group). Nevertheless, the smooth seeds, as a whole, showed the lowest values of solidity in their dorsal views due to the wide seed dorsal morphologies. Most of the species corresponded to the type denominated as dorso-canaliculata by Boissier and Rohrbach [1,2], having wings or channel-like structures that produce entrants or depressions in the upper and lower parts of their dorsal seed view. In addition, some of the species also showed situate margins. In convex figures, solidity reaches the maximum value of 1. In fact, a kind of gradient was observed between the three seed groups based on the dorsal part since the concavities or depressions were more apparent in smooth seeds, intermediate in rugose seeds and absent in the seeds with spines that they have constituted the group of convex seeds and hence the seeds with the maximum solidity values. In summary, we have two extreme positions formed by smooth and echinate seeds that corresponded to the types dorso-canaliculata and dorso-plana following the traditional terminology [1,2].

Convex figures have reduced values of the perimeter/area ratio in comparison with their non-convex counterparts. The perimeter/area relationship in plane figures represents an estimation of the surface/volume ratio, a measurement important in living organisms in general, and, in particular, for seeds. The surface/area ratio has a role in the regulation of metabolic aspects such as nutrient availability and gas exchange, including evaporation and oxygen availability. Given the importance of this ratio, we would suppose that solidity is an important property of seeds, and this feature should be considered for taxonomic plant studies.

The infrageneric taxonomic groups based on DNA sequencing analysis [3] and the obtained morphological seed groups showed a notable relationship. The majority of the species from the group of smooth and rugose seeds belonged to *S.* subg. *Silene*, whereas the group of echinate seeds to *S.* subg. *Behenantha*. The subgenus *S.* subg. *Lychnis* did not show any studied sample within the group of smooth seeds, but most of the studied species had echinate seeds. For certain sections of *S.* subg. *Silene* and *S.* subg. *Behenantha*, this morphological approach also supported their current taxonomical identification. Most of the smooth seeds belonged to *S.* sect. *Silene*, although a low number of species were also characterized by rugose and echinate seeds. Similarly, the species of *S.* sect. *Siphonomorpha* showed the three types of seeds, but with a clear dominance by any of them. The notable dominance of species of *S.* subg. *Behenantha* in the group of echinate seeds was correlated with the presence of all studied species of S. sect. *Melandrium* (*S. diclinis*, *S. latifolia* and *S. diclinis*) only within this group. These species, together with the dioecious taxa *S. heuffelii* Soó and *S. marizii* Samp. form a strongly supported monophyletic group [34], which might be supported by these morphological seed features. Recent morphological studied based on SEM images also supported the taxonomical identification of this section [4]. The species *S. viscosa* (*S.* subg. *Behenantha* sect. *Physolychnis*) has been reported to form hybrids with *S. latifolia* (*S.* subg. *Behenantha* sect. *Melandrium*) [35], supporting a close relation between both sections. Nevertheless, a further application of this seed classification would be helpful for the infrageneric taxonomical classification of this complex and diverse genus, though more species of the different subgenera and sections should be added for supporting the here obtained data. As Hoseini et al. [4] stated, designating credible morphological features needs a wider range of data and species to cover.

## 5. Conclusions

The seeds of *Silene* species were classified in three groups according to their lateral silhouette, named as smooth, rugose and echinate. Differences between the groups were found in the majority of the studied quantitative morphological characters, including area, perimeter, circularity, roundness and solidity. Solidity is an interesting property related to the degree of convexity of a plane surface. Among the characteristics measured in seeds, solidity is the most conserved. No differences were found in solidity values between the three groups for the lateral view of the seeds, but differences were found in the dorsal views. The three groups are related to the infrageneric taxonomic groups based on DNA sequence analysis. The species of the group of smooth seeds mostly belonged to *S.* subg. *Silene*, and specifically to *S.* sect. *Silene*, whereas the species of the group of echinate seeds mainly corresponded to *S.* subg. *Behenantha*. This group contains all the species tested of *S.* sect. *Melandrium* and related species, as well as most of the species of *S.* subg. *Lychnis*. In the case of the group of rugose seeds, the majority of species belong to *S.* subg. *Silene*, and a high variety of members of different sections were included, but the most common ones corresponded to *S.* sect. *Siphonomorpha* and *S.* sect. *Silene*.

**Author Contributions:** Conceptualization, E.C.; data curation, J.J.M.-G.; formal analysis, J.J.M.-G., J.L.R.-L., A.J., Á.T., B.J. and E.C.; investigation, J.J.M.-G., J.L.R.-L., A.J., Á.T., B.J. and E.C.; methodology, J.L.R.-L., A.J., J.J.M.-G., Á.T. and E.C.; resources, J.J.M.-G., J.L.R.-L., A.J., Á.T., B.J. and E.C.; software, J.J.M.-G. and E.C.; validation, J.L.R.-L., A.J., Á.T., B.J. and E.C.; visualization, J.J.M.-G., J.L.R.-L., A.J., B.J. and E.C.; writing—original draft, E.C.; writing—review and editing, J.J.M.-G., J.L.R.-L., A.J., Á.T., B.J. and E.C. All authors have read and agreed to the published version of the manuscript.

**Funding:** Project "CLU-2019-05-IRNASA/CSIC Unit of Excellence", funded by the Junta de Castilla y León and co-financed by the European Union (ERDF "Europe drives our growth").

**Institutional Review Board Statement:** Not applicable.

**Informed Consent Statement:** Not applicable.

**Data Availability Statement:** The raw data used in the general morphological analysis are stored in Zenodo (https://zenodo.org/deposit/6772881; accessed on 15 June 2022).

**Conflicts of Interest:** The authors declare no conflict of interest.

## Appendix A

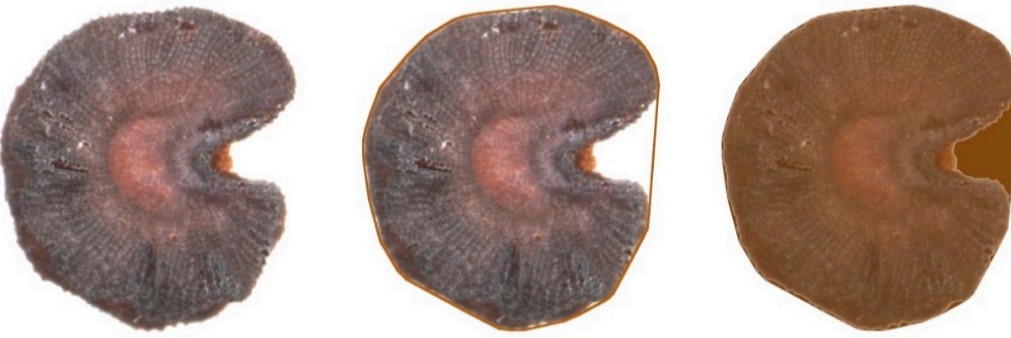

**Figure A1.** A seed of *Silene apetala* (**left**) with the corresponding convex hull (**center**). The image on the (**right**) shows the area of the convex hull corresponding to the seed image. Solidity is the ratio between the area of an object and the area of a convex hull of the same object. In the seed image of the figure, solidity equals 0.94. Solidity equals one in convex figures.

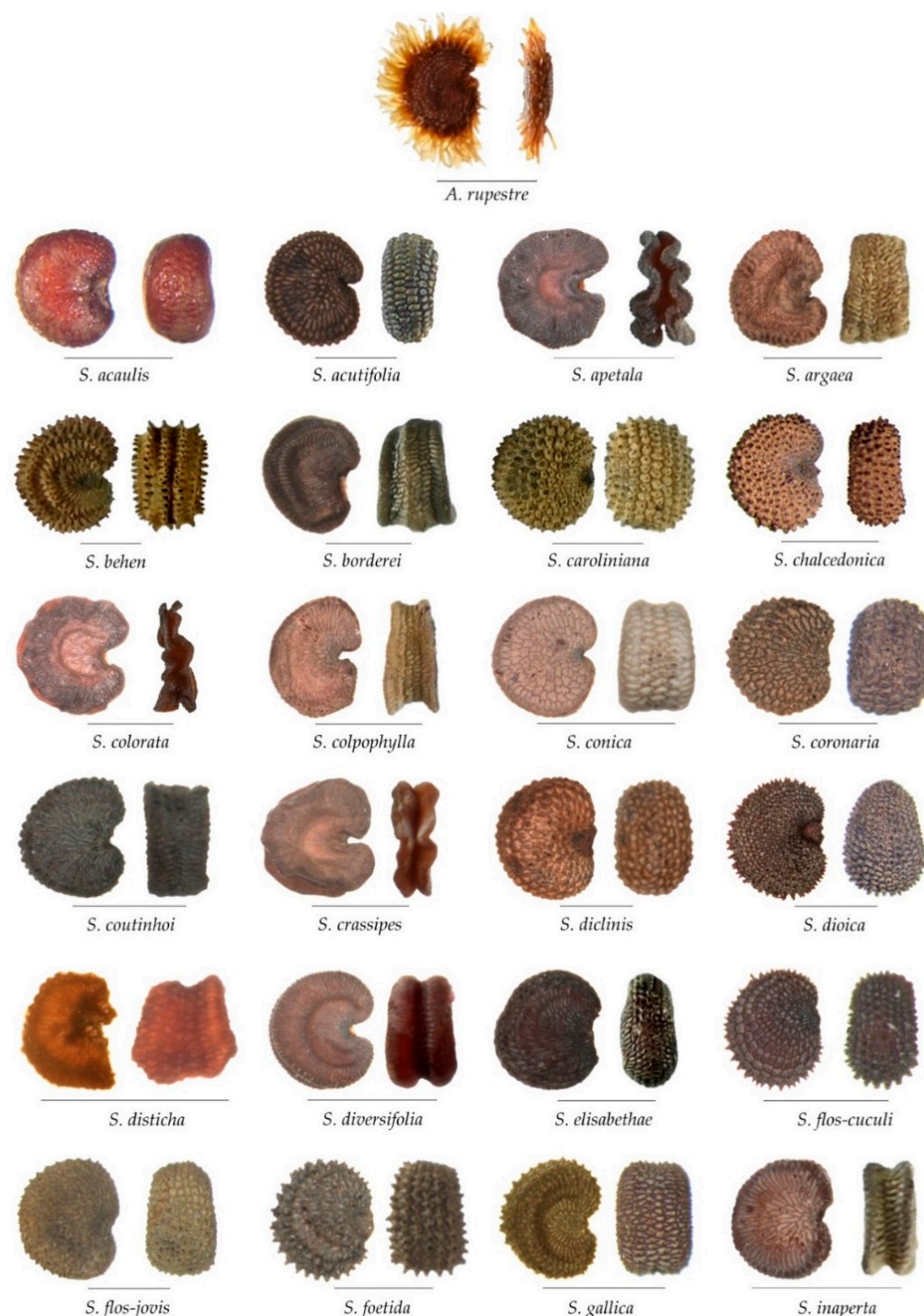

**Figure A2.** *Cont.*

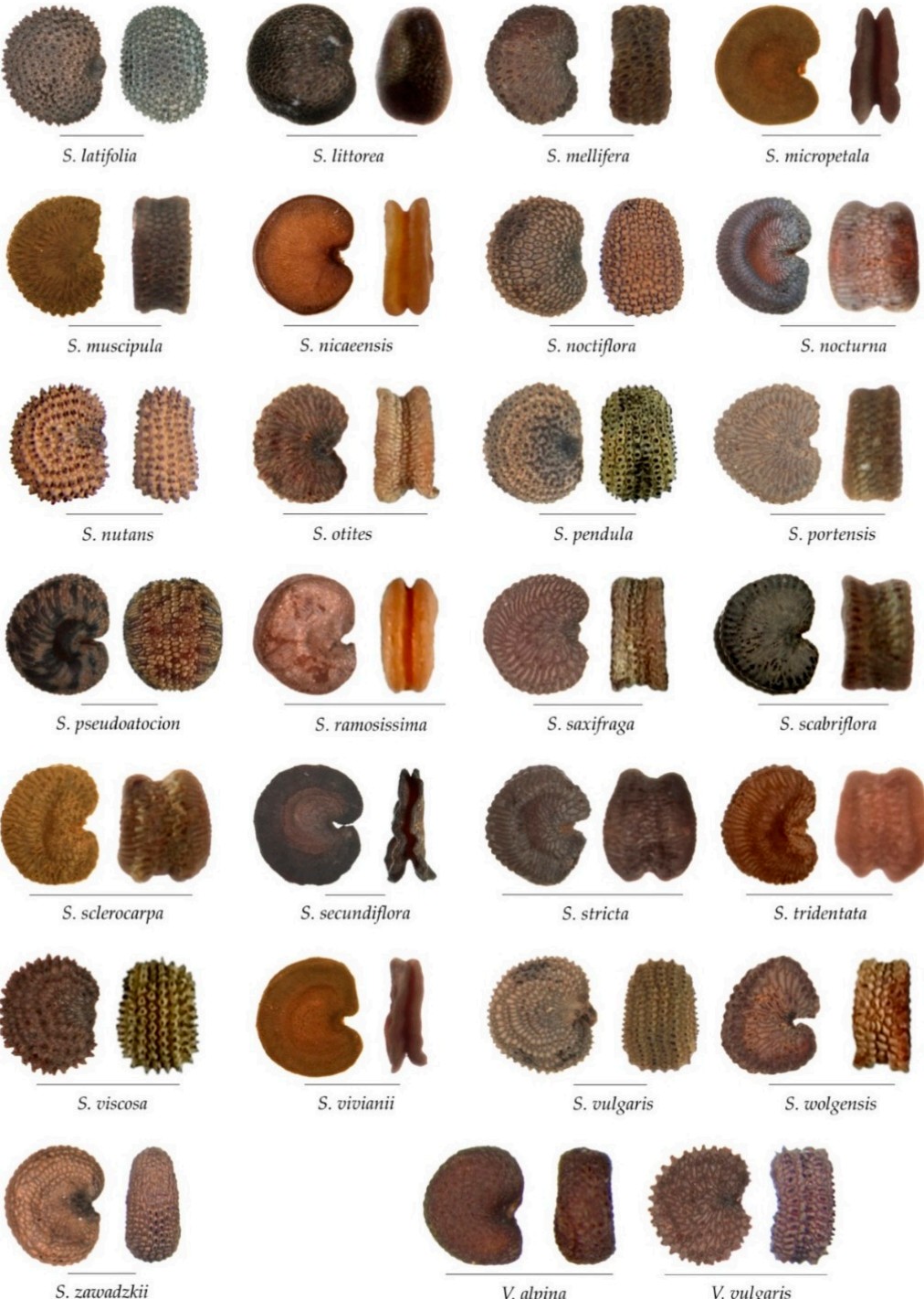

**Figure A2.** Representative photographs of the lateral and dorsal views of the seeds for the species of *Atocion (A.)*, *Silene (S.)* and *Viscaria (V.)* used in this work. Bars represent 1 mm.

## Appendix B

**Table A1.** A total of 52 species and 104 populations belonging to *Silene*, *Atocion* and *Viscaria* are listed with an indication of the origin of the samples. The life span (annual, biannual and perennial plants) is indicated for each species according to Talavera [36], Morton [37], Chater and Walters [38], Davis et al. [39] and Danin and Fragman [40]. The taxonomical assignment of each species to subgenera and sections is taken from [3,41]. In the column, origin u. means unknown.

| Species | Origin | Annual/Biannual/Perennial | Subgenus and Section |
|---|---|---|---|
| *S. acaulis* (L.) Jacq. | Botanical Garden of Babeș-Bolyai University. Cluj, Romania | Perennial | *S.* subg. *Silene* sect. *Siphonomorpha* Otth |
| *S. acaulis* (L.) Jacq | University of Oulu, Finland | Perennial | *S.* subg. *Silene* sect. *Siphonomorpha* Otth |
| *S. acutifolia* Link ex Rohrb. | Czech Republic | Perennial | *S.* subg. *Behenantha* (Otth) Torr. & A.Gray sect. *Acutifoliae* Oxelman & F.Jafari |
| *S. apetala* Willd. | Arenal de Petrer, Alicante, Spain | Annual | *S.* subg. *Silene* sect. *Silene* |
| *S. apetala* Willd. | Caprala, Petrer, Alicante, Spain | Annual | *S.* subg. *Silene* sect. *Silene* |
| *S. apetala* Willd. | u. (IBP collection, Brno) | Annual | *S.* subg. *Silene* sect. *Silene* |
| *S. argaea* Fisch. & C.A.Mey. | u. (IBP collection, Brno) | Perennial | *S.* subg. *Silene* sect. *Auriculatae* (Boiss.) Schischk. |
| *S. behen* L. | u. (IBP collection, Brno) | Annual | *S.* subg. *Behenantha* (Otth) Torr. & A.Gray sect. *Behenantha* Otth |
| *S. borderei* Jord. | u. (IBP collection, Brno) | Perennial | *S.* subg. *Silene* sect. *Silene* |
| *S. caroliniana* Walter | u. (IBP collection, Brno) | Perennial | *S.* subg. *Behenantha* (Otth) Torr. & A.Gray sect. *Physolychnis* (Benth.) Bocquet |
| *S. chalcedonica* (L.) E.H.L.Krause | u. (IBP collection, Brno) | Perennial | *S.* subg. *Lychnis* (L.) Greuter sect. *Lychnis* (L.) Greuter |
| *S. colorata* Poir. | Arenal de Petrer, Alicante, Spain | Annual | *S.* subg. *Silene* sect. *Silene* |
| *S. colorata* Poir. | Villena, Alicante, Spain | Annual | *S.* subg. *Silene* sect. *Silene* |
| *S. colorata* Poir. | u. (IBP collection, Brno) | Annual | *S.* subg. *Silene* sect. *Silene* |
| *S. colpophylla* Wrigley | France | Perennial | *S.* subg. *Silene* sect. *Siphonomorpha* Otth |
| *S. colpophylla* Wrigley | u. (IBP collection, Brno) | Perennial | *S.* subg. *Silene* sect. *Siphonomorpha* Otth |
| *S. conica* L. | Villena Alicante, Spain | Annual | *S.* subg. *Behenantha* (Otth) Torr. & A.Gray sect. *Conoimorpha* Otth |
| *S. conica* L. | Albarracín, Teruel, Spain | Annual | *S.* subg. *Behenantha* (Otth) Torr. & A.Gray sect. *Conoimorpha* Otth |
| *S. conica* L. | Berlin | Annual | *S.* subg. *Behenantha* (Otth) Torr. & A.Gray sect. *Conoimorpha* Otth |

**Table A1.** *Cont.*

| Species | Origin | Annual/Biannual/Perennial | Subgenus and Section |
|---|---|---|---|
| *S. conica* L. | Berlin | Annual | *S.* subg. *Behenantha* (Otth) Torr. & A.Gray sect. *Conoimorpha* Otth |
| *S. coronaria* (L.) Clairv. | Rostock Botanical garden | Perennial | *S.* subg. *Lychnis* (L.) Greuter sect. *Agrostemma* (DC.) Greuter |
| *S. coronaria* (L.) Clairv. | University of Berlin | Perennial | *S.* subg. *Lychnis* (L.) Greuter sect. *Agrostemma* (DC.) Greuter |
| *S. coutinhoi* Rothm. & P.Silva | Larouco, Ourense, Spain | Perennial | *S.* subg. *Silene* sect. *Siphonomorpha* Otth |
| *S. crassipes* Fenzl | Hania, Kriti, Greece | Annual | *S.* subg. *Silene* sect. *Lasiocalycinae* (Boiss.) Chowdhuri |
| *S. diclinis* (Lag.) M.Laínz | u. (IBP collection, Brno) | Perennial | *S.* subg. *Behenantha* (Otth) Torr. & A.Gray sect. *Melandrium* (Röhl.) Rabeler |
| *S. diclinis* (Lag.) M.Laínz | u. (IBP collection, Brno) | Perennial | *S.* subg. *Behenantha* (Otth) Torr. & A.Gray sect. *Melandrium* (Röhl.) Rabeler |
| *S. diclinis* (Lag.) M.Laínz | u. (IBP collection, Brno) | Perennial | *S.* subg. *Behenantha* (Otth) Torr. & A.Gray sect. *Melandrium* (Röhl.) Rabeler |
| *S. diclinis* (Lag.) M.Laínz | Pla de Mora | Perennial | *S.* subg. *Behenantha* (Otth) Torr. & A.Gray sect. *Melandrium* (Röhl.) Rabeler |
| *S. dioica* (L.) Clairv. | Berlin | Perennial | *S.* subg. *Behenantha* (Otth) Torr. & A.Gray sect. *Melandrium* (Röhl.) Rabeler |
| *S. dioica* (L.) Clairv. | Rostock Botanical garden | Perennial | *S.* subg. *Behenantha* (Otth) Torr. & A.Gray sect. *Melandrium* (Röhl.) Rabeler |
| *S. dioica* (L.) Clairv. | Orto Botanico Friulano, Udine Italy | Perennial | *S.* subg. *Behenantha* (Otth) Torr. & A.Gray sect. *Melandrium* (Röhl.) Rabeler |
| *S. disticha* Willd. | Alcari, Estremadura, Portugal | Annual | *S.* subg. *Silene* sect. *Silene* |
| *S. diversifolia* Otth | Elda, Alicante, Spain | Annual | *S.* subg. *Silene* sect. *Silene* |
| *S. elisabethae* Jan | u. (IBP collection, Brno) | Perennial | *S.* subg. *Behenantha* (Otth) Torr. & A.Gray unplaced |
| *S. flos-cuculi* (L.) Greuter & Burdet | Berlin | Perennial | *S.* subg. *Lychnis* (L.) Greuter sect. *Coccyganthe* (Rchb.) Greuter |
| *S. flos-jovis* (L.) Greuter & Burdet | Berlin | Perennial | *S.* subg. *Lychnis* (L.) Greuter sect. *Agrostemma* (DC.) Greuter |
| *S. flos-jovis* (L.) Greuter & Burdet | Gogela St Gallen | Perennial | *S.* subg. *Lychnis* (L.) Greuter sect. *Agrostemma* (DC.) Greuter |

**Table A1.** *Cont.*

| Species | Origin | Annual/Biannual/Perennial | Subgenus and Section |
|---|---|---|---|
| *S. foetida* Link ex Spreng. | Muiños, Ourense, Spain | Perennial | *S.* subg. *Behenantha* (Otth) Torr. & A.Gray sect. *Acutifoliae* Oxelman & F.Jafari |
| *S. gallica* L. | Corse, France | Annual | *S.* subg. *Silene* sect. *Silene* |
| *S. gallica* L. | u. (IBP collection, Brno) | Annual | *S.* subg. *Silene* sect. *Silene* |
| *S. gallica* L. | St. Gallen, Switzerland | Annual | *S.* subg. *Silene* sect. *Silene* |
| *S. inaperta* L. | Elda, Alicante, Spain | Annual | *S.* subg. *Silene* sect. *Muscipula* |
| *S. inaperta* L. | Petrer, Alicante, Spain | Annual | *S.* subg. *Silene* sect. *Muscipula* (Tzvelev) Oxelman, F.Jafari & Gholipour |
| *S. inaperta* L. | u. (IBP collection, Brno) | Annual | *S.* subg. *Silene* sect. *Muscipula* (Tzvelev) Oxelman, F.Jafari & Gholipour |
| *S. latifolia* Poir. | Pego Alicante, Spain | Perennial | *S.* subg. *Behenantha* (Otth) Torr. & A.Gray sect. *Melandrium* (Röhl.) Rabeler |
| *S. latifolia* Poir. | Brno, neighborhood of Bystrc, South Moravia. CZ | Perennial | *S.* subg. *Behenantha* (Otth) Torr. & A.Gray sect. *Melandrium* (Röhl.) Rabeler |
| *S. latifolia* Poir. | Brno, neighborhood of Bystrc, South Moravia. CZ | Perennial | *S.* subg. *Behenantha* (Otth) Torr. & A.Gray sect. *Melandrium* (Röhl.) Rabeler |
| *S. latifolia* Poir. | Larzac_Fr_2019 | Perennial | *S.* subg. *Behenantha* (Otth) Torr. & A.Gray sect. *Melandrium* (Röhl.) Rabeler |
| *S. latifolia* Poir. | Locality of Liblice, district of Mělník, Central Bohemia. CZ | Perennial | *S.* subg. *Behenantha* (Otth) Torr. & A.Gray sect. *Melandrium* (Röhl.) Rabeler |
| *S. latifolia* Poir. | Internal laboratoy material BFU. Selfcross (17 generations) | Perennial | *S.* subg. *Behenantha* (Otth) Torr. & A.Gray sect. *Melandrium* (Röhl.) Rabeler |
| *S. littorea* Brot. | Melides, Baixo Alentejo, Portugal | Annual | *S.* subg. *Behenantha* (Otth) Torr. & A.Gray sect. *Psammophilae* (Talavera) Greuter |
| *S. mellifera* Boiss. & Reut. | Calatayud, Zaragoza, Spain | Perennial | *S.* subg. *Silene* sect. *Siphonomorpha* Otth |
| *S. micropetala* Lag. | Nador, Morocco | Annual | *S.* subg. *Silene* sect. *Silene* |
| *S. muscipula* L. | Granada, Spain | Annual | *S.* subg. *Silene* sect. *Muscipula* (Tzvelev) Oxelman, F.Jafari & Gholipour |
| *S. nicaeensis* All. | Benicarló, Castellón, Spain | Biannual/Perennial | *S.* subg. *Silene* sect. *Silene* |
| *S. nicaeensis* All. | Algarve, Portugal | Biannual/Perennial | *S.* subg. *Silene* sect. *Silene* |
| *S. noctiflora* L. | Kuřim (CzR) | Annual | *S.* subg. *Behenantha* (Otth) Torr. & A.Gray sect. *Elisanthe* (Fenzl ex Endl.) Ledeb. |
| *S. noctiflora* L. | St Gallen, Switzerland | Annual | *S.* subg. *Behenantha* (Otth) Torr. & A.Gray sect. *Elisanthe* (Fenzl ex Endl.) Ledeb. |

**Table A1.** *Cont.*

| Species | Origin | Annual/Biannual/Perennial | Subgenus and Section |
|---|---|---|---|
| *S. noctiflora* L. | St Gallen, Switzerland, 1986 | Annual | *S.* subg. *Behenantha* (Otth) Torr. & A.Gray sect. *Elisanthe* (Fenzl ex Endl.) Ledeb. |
| *S. nocturna* L. | L'Abdet Alicante, Spain | Annual | *S.* subg. *Silene* sect. *Silene* |
| *S. nocturna* L. | Villena, Alicante, Spain | Annual | *S.* subg. *Silene* sect. *Silene* |
| *S. nocturna* L. | Peñíscola, Castellón, Spain | Annual | *S.* subg. *Silene* sect. *Silene* |
| *S. nocturna* L. | Forcall, Castellón, Spain | Annual | *S.* subg. *Silene* sect. *Silene* |
| *S. nocturna* L. | Czech Republic | Annual | *S.* subg. *Silene* sect. *Silene* |
| *S. nutans* L. | Besançon, France | Perennial | *S.* subg. *Silene* sect. *Siphonomorpha* Otth |
| *S. nutans* L. | Rostok, Germany. 2003. | Perennial | *S.* subg. *Silene* sect. *Siphonomorpha* Otth |
| *S. otites* (L.) Wibel | u. (IBP collection, Brno) | Biannual/Perennial | *S.* subg. *Silene* sect. *Siphonomorpha* Otth |
| *S. otites* (L.) Wibel | u. (IBP collection, Brno) | Biannual/Perennial | *S.* subg. *Silene* sect. *Siphonomorpha* Otth |
| *S. otites* (L.) Wibel | Bot.Al.Borza_Romania_2000 | Biannual/Perennial | *S.* subg. *Silene* sect. *Siphonomorpha* Otth |
| *S. pendula* L. | Czech Republic | Annual | *S.* subg. *Behenantha* (Otth) Torr. & A.Gray sect. *Behenantha* Otth |
| *S. portensis* L. | Gea, Teruel, Spain | Annual | *S.* subg. *Silene* sect. *Portenses* F.Jafari & Oxelman |
| *S. pseudoatocion* Desf. | Torretes, Ibi, Alicante | Annual | *S.* subg. *Silene* sect. *Silene* |
| *S. ramosissima* Desf. | Oliva, Valencia, Spain | Annual | *S.* subg. *Silene* sect. *Siphonomorpha* Otth |
| *S. saxifraga* L. | u. (IBP collection, Brno) | Perennial | *S.* subg. *Silene* sect. *Siphonomorpha* Otth |
| *S. scabriflora* Brot. | Cavado, Minho, Portugal | Annual | *S.* subg. *Silene* sect. *Silene* |
| *S. sclerocarpa* Dufour | Villena, Alicante, Spain | Annual | *S.* subg. *Silene* sect. *Silene* |
| *S. sclerocarpa* Dufour | Peñíscola, Castellón, Spain | Annual | *S.* subg. *Silene* sect. *Silene* |
| *S. sclerocarpa* Dufour | Nador, Morocco | Annual | *S.* subg. *Silene* sect. *Silene* |
| *S. secundiflora* Otth | Pego Alicante, Spain | Annual | *S.* subg. *Silene* sect. *Silene* |
| *S. stricta* L. | Barbate, Cádiz, Spain | Annual | *S.* subg. *Silene* sect. *Muscipula* (Tzvelev) Oxelman, F.Jafari & Gholipour |
| *S. tridentata* Desf. | Villena, Alicante, Spain | Annual | *S.* subg. *Silene* sect. *Silene* |
| *S. tridentata* Desf. | Peñíscola, Castellón, Spain | Annual | *S.* subg. *Silene* sect. *Silene* |
| *S. viscosa* (L.) Pers. | u. (IBP collection, Brno) | Biannual/Perennial | *S.* subg. *Behenantha* (Otth) Torr. & A.Gray sect. *Physolychnis* (Benth.) Bocquet |
| *S. viscosa* (L.) Pers | u. (IBP collection, Brno) | Biannual/Perennial | *S.* subg. *Behenantha* (Otth) Torr. & A.Gray sect. *Physolychnis* (Benth.) Bocquet |
| *S. vivianii* Steud. | Saidia, Morocco | Annual | *S.* subg. *Silene* sect. *Silene* |

**Table A1.** *Cont.*

| Species | Origin | Annual/Biannual/Perennial | Subgenus and Section |
|---|---|---|---|
| *S. vulgaris* (Moench) Garcke | Elda, Alicante, Spain | Perennial | *S.* subg. *Behenantha* (Otth) Torr. & A.Gray sect. *Behenantha* Otth |
| *S. vulgaris* (Moench) Garcke | Forcall, Castellón, Spain | Perennial | *S.* subg. *Behenantha* (Otth) Torr. & A.Gray sect. *Behenantha* Otth |
| *S. vulgaris* (Moench) Garcke | CzR | Perennial | *S.* subg. *Behenantha* (Otth) Torr. & A.Gray sect. *Behenantha* Otth |
| *S. wolgensis* (Hornem.) Otth | Bashkortostan, Rusia. | Biannual/Perennial | *S.* subg. *Silene* sect. *Siphonomorpha* Otth |
| *S. zawadzkii* Herbich | Berlin, Germany | Perennial | *S.* subg. *Behenantha* (Otth) Torr. & A.Gray sect. *Physolychnis* (Benth.) Bocquet |
| *A. rupestre* (L.) Oxelman | u. (IBP collection, Brno) | Perennial | - |
| *V. alpina* (L.) G.Don | Linz | Perennial | - |
| *V. vulgaris* Röhl. | u. (IBP collection, Brno) | Perennial | - |
| *V. vulgaris* Röhl. | Klembów PL 1951r. OBUW | Perennial | - |
| *V. vulgaris* Röhl. | Berlin | Perennial | - |

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
