# Peer review of "Seed Morphological Properties Related to Taxonomy in Silene L. Species"

_2673-6500, doi:10.3390/taxonomy2030024_

Round 1

Reviewer 1 Report

Dear Authors, I congratulate you on an interesting and correctly prepared and performed research. However, before final acceptance of the manuscript for publication, please correct and/or pay attention to the following few notes: 1.    The author of the genus Silene (i.e. Silene L.) should be mentioned either in the title or in the Introduction (at the first mention of the genus). 2.    The name of the author of the family should also be mentioned in the introduction. 3.    Also, at the first mention or in the Table 1, the authors of the subgenus and sections should be added. 4.    In the keywords: (i) DO NOT REPEAT words from the title (e.g. taxonomy); (ii) reorganize words related to features of seeds - it is better to use terms (e.g. maybe not all the listed words are necessary because they have too many meanings, like "area", but the term "solidity of seeds" is missing); (iii) it would be desirable to add the word Caryophyllaceae (if you will be able to manage the number of keywords). 5.    In the Material and methods chapter, in subsection 2.1. it should be stated according to which reference the species nomenclature is adopted (I assume according to the WCVP?). Alternatively, it can be added into the Table A1 - immediately comment on it: In subsection 2.1. authors stated that Table A1 is in Appendix 1, but at the end of the manuscript the statement is that it is in the Appendix B. Please check and make sure to match it! Additional comment: In the title of Table A1, it is not necessary to list the names of the authors along with the reference numbers. However, those numbers are not consistent with the reference list, so please, see my comment about the reference list. 6.    If the photos of the seeds in Figure 1 are the results of this research (and it seems they are), then they should be in the Results chapter or just mentioned in that chapter, and shown in the Appendix. In addition, in the title of Figure 1, next to the names of the genera, their abbreviations, which are used below the photos, should be added. Also, for better visibility, it would be better to separate Fig. 1 into two plates or figures. 7.    VERY IMPORTANT NOTE that may indicate SELF-PLAGIARISM: the description of the methods in this manuscript and in the work of the same authors (Taxonomy 20222(1), 1-19; https://doi.org/10.3390/taxonomy2010001; by the way, in this manuscript, that paper is cited by wrong volume!) are too similar and some parts are repeated in this manuscript / including the similar drawings shown in Fig. 2). This SHOULD be avoided by quoting parts in the Methods of this manuscript from a previous paper! Or just quote a previous paper…! 8.    In the list of references, there was a shift in the serial numbers of references 16-17 and 31-32, which caused a discrepancy in the citation of the literature in the text and the list (cf. my note 4).   Best wishes in your future work!

Author Response

Dear Reviewer,

Thank you very much for your commentaries that have contributed notably to improve the quality of the article. Please find below our responses to them:

  1. The author of the genus Silene is now mentioned in the title of the article.
  2. The name of the author of the family Caryophyllaceae is now mentioned in the introduction.
  3. The authors of the subgenus and sections have been added in Table 1.
  4. Words that were repeated in the keywords and in the title were eliminated from the list of keywords; also, words have been reorganized and the terms “solidity of seeds” and “Caryophyllaceae” have been added.
  5. In the Materials and methods section, it is now indicated that plant nomenclature and the corresponding authorities for each taxa are written according to POWO and Jafari et al. (2020). The situation of Table B1 in Appendix B has been corrected.
  6. The photos of seeds are now presented in the results section. The abbreviations to the names of the genera used in Figure 1 are now given in the title.
  7. The description of the Methods section has been modified, avoiding the similarity in some sections with our previous article in Taxonomy. The Figure 2 (now Figure 1) was originally made for this article.
  8. The reference list has been corrected.

Thank you very much for your commentaries to the article. With best regards,

Emilio Cervantes

Reviewer 2 Report

I suggest to review all the number of references in brackets along the text, because many inconsistencies have been found.

Besides, you must explain why many samples came from unknown origin.

Author Response

Dear Reviewer,

Thank you very much for your review that has contributed to improve the quality of the article. Please find below our responses to your commentaries:

  • The references and reference list have been corrected as indicated
  • The title of the article in ref 19 (now ref. 21) has been corrected.
  • Text in lines 51-52 has been changed to:

In relation to general seed shape, the description is based on terms, such as circular, globose, reniform, piriform, but lacks quantitative measurements.

  • The mention in the text corresponding to Table B1 in Appendix B has been corrected.
  • The table indicates now to which collection belong the seeds labelled as of unknown origin.
  • The terms interrupted by hyphens have been revised and corrected as well as other terms indicated(“papillose”).

Thank you very much for your commentaries to the article. With best regards,

Emilio Cervantes

Reviewer 3 Report

The paper is well written and provides an important contribution to the morphology of the seeds of  the genus Silene, also evaluated with statistical methods. The number of examined species is conspicuous and there is a good correspondence with the groups identified on the basis of other characters and DNA.

the work can therefore be accepted in this form, I will limit myself only to highlighting some typos and little suggestions:

page 2 line 96 some more references could be added for example:   Brullo et al. (2014). Silene peloritana (Caryophyllaceae) a new species from Sicily. PHYTOTAXA 172:256-264, doi: http://dx.doi.org/10.11646/phytotaxa.172.3.6

page 3 line 115 page 6 line 133 check the exact correspondence of the numbers [22] [23] with the references because it seems to me that they do not correspond

page  5 line 129 put Atocion Silene and Viscaria in italics

page 25 line 595 insert the year 2022 in this reference

Author Response

Dear Reviewer,

Thank you very much for your commentaries that have contributed to improve the quality of the article.

The article suggested as a new reference has been added as Ref. 18. It is mentioned in the text as:

Given the complexity and broad geographic distribution of this genus the work has been focused on a range of diverse floras, such as for example those of Egypt [8], Iran [10-12], Jordan [13], Korea [14] Pakistan [15], Spain [16], the Mediterranean area [17,18] or Turkey [19].

The references have been reviewed and the reference list, corrected.

All other corrections in the text have been done as indicated.

Thank you very much for your positive commentaries to the article. With best regards,

Emilio Cervantes

Reviewer 4 Report

Dear Authors, necessary explanations are written on the manuscript.

A good scientific path has been followed in the article, and the aim and the result are compatible with each other. Some errors have been identified in the references given in the article. All references in the article need to be rechecked.

Author Response

Dear Reviewer,

Thank you very much for your commentaries that have contributed to improve the quality of the article. Please find below our responses to them:

  • References have been rechecked and corrected as indicated
  • Year in Campbell and Skillings (1985) [31] has been deleted.

With best regards,

Emilio Cervantes

Reviewer 5 Report

I found the manuscript interesting on the basis of the merits following:

1-     The manuscript is well-written and have a good readability.

2-     The subject is considerably novel, especially for applying new criteria and terms for describing morphological features of seeds in the genus Silene.

3-     The manuscript provided a good methodology which could be reproduce and apply for the other species of Silene and therefore could help to the identification and classification of the species of the genus.

4-     The main question of the manuscript was whether seed features could contribute to the classification of the genus Silene, which I found acceptable and well addressed.

5-     The manuscripts covered a good variation of the species of Silene and its related genera (52 species) which allowed the authors to have a fair conclusion.

6-     The data provided could be applied for further analysis, especially evolutionary and phylogenetic approaches.

7-     I found the conclusion consistent with the evidence and arguments provided.

Therefore, I recommend the publication of the manuscript.

Author Response

Dear Reviewer,

Thank you very much for your positive commentaries to the article. We are satisfied to see that our work is recognized. With compliments,

Emilio Cervantes
